# Demystifying Hybrid Thinking: Can LLMs Truly Switch Between Think and No-Think?

## Abstract

Hybrid thinking enables LLMs to switch between reasoning and direct answering, offering a balance between efficiency and reasoning capability. Yet our experiments reveal that current hybrid thinking LLMs only achieve partial mode separation: reasoning behaviors often leak into the no-think mode. To understand and mitigate this, we analyze the factors influencing controllability and identify four that matter most: (1) larger data scale, (2) using think and no-think answers from different questions rather than the same question, (3) a moderate increase in no-think data number, and (4) a two-phase strategy that first trains reasoning ability and then applies hybrid think training. Building on these findings, we propose a practical recipe that, compared to standard training, can maintain accuracy in both modes while significantly reducing no-think output length (from 1085 to 585 on MATH500) and occurrences of reasoning-supportive tokens such as "wait" (from 5917 to 522 on MATH500). Our findings highlight the limitations of current hybrid thinking and offer directions for strengthening its controllability. The code is available at: https://anonymous.4open.science/r/hybrid-thinking-iclr2026EBEF..

## 1 Introduction

> **Question:** Please reason step by step and put your final answer within boxed{}. Let triangle $ABC$ be inscribed in a circle $\omega$. The tangents to $\omega$ at $B$ and $C$ intersect at $D$. Let the line $\overline{AD}$ meet $\omega$ again at $P$. Given $AB = 5$, $BC = 9$, and $AC = 10$, suppose $AP = \frac{m}{n}$ in lowest terms. Find $m + n$.
>
> Final Answer: 113

> **NO_THINK:**
> <think>\n\n</think>\n\nWe are given triangle $ABC$ inscribed in a circle $\omega$ ... Step 1: ... **Wait** — this is not correct ... Final Answer: 51

> **THINK:**
> <think>\nOkay, so I need to solve this geometry problem. Let me try to understand it step by step ... First ... Hmm ... **Wait**, ... Thus, AP = 100/13, and since m = 100 and n = 13 are relatively prime, we compute m + n = 100 + 13 = 113

Figure 1: Case study on an AIME24 problem. We compare the responses of Qwen3-8B under no-think and think modes. In the no-think mode, Qwen3-8B still performs reasoning outside the no-think constraint (e.g., generating reasoning-supportive words such as `Wait`), indicating that its hybrid thinking ability remains imperfect and cannot achieve full control.

Hybrid thinking (Sui et al., 2025; Chen et al., 2024b) is a widely adopted approach in many company models, enabling control over whether the model engages in reasoning and thereby achieving a more efficient and flexible reasoning process. Examples include Gemini (Team et al., 2025a), GPT-oss (Agarwal et al., 2025), Qwen3 (Yang et al., 2025), and DeepSeek V3.1 (Liu et al., 2024). However, few studies have systematically investigated the capabilities of hybrid thinking models, such as the training factors that influence them and the limitations they face in practice. This paper fills this gap by 1) offering a comprehensive analysis of hybrid thinking and 2) introducing a practical training recipe to enhance its controllability.

We begin by exploring the output differences of hybrid thinking models. Hybrid thinking is typically implemented by adding control tokens (e.g., \no_think, \think) into the prompt and applying supervised fine-tuning (SFT). Evaluating Qwen3-8B on MATH500, AIME24 and GPQA, with Qwen2.5-7B-Instruct as a pure no-think baseline, we find that hybrid models perform better in the

Table 1: Performance on MATH500, AIME24, and GPQA under think and no-think modes. We report accuracy (Acc.), output length (Len.), #Wait count (i.e., occurrences of reflection words such as "wait", "hmm", "`alternatively`"), and **no-think differences** ($\Delta$) between Hybrid (Qwen3-8B) and Instruct (Qwen2.5-7B-Instruct). Notably, Hybrid model in the no-think mode produce much longer outputs than the Instruct model, and even generate reflection words.

| Model | Acc. (%) | | | Len. | | | #Wait | | |
|---|---|---|---|---|---|---|---|---|---|
| | Think | No-think | $\Delta$ | Think | No-think | $\Delta$ | Think | No-think | $\Delta$ |
| MATH500 | | | | | | | | | |
| Instruct | – | 59.94 | | – | 703.11 | | – | 0 | |
| Hybrid | 92.82 | 82.90 | **+22.96** | 4384.35 | 958.31 | **+255.20** | 83703 | 646 | **+646** |
| AIME24 | | | | | | | | | |
| Instruct | – | 6.67 | | – | 1729.22 | | – | 0 | |
| Hybrid | 63.33 | 24.00 | **+17.33** | 11394.54 | 4061.67 | **+2332.45** | 12184 | 184 | **+184** |
| GPQA | | | | | | | | | |
| Instruct | – | 30.15 | | – | 775.07 | | – | 0 | |
| Hybrid | 59.14 | 47.93 | **+17.78** | 7451.07 | 1364.96 | **+589.89** | 66152 | 616 | **+616** |

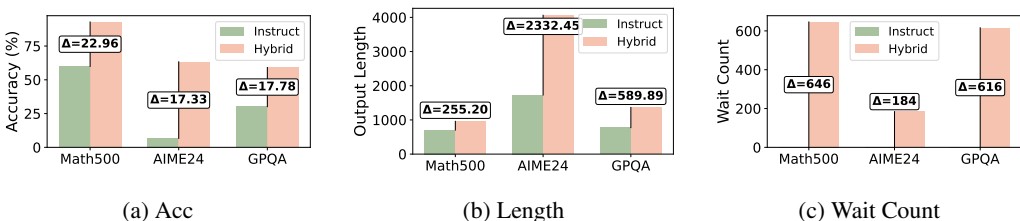

| (a) Acc | (b) Length | (c) Wait Count |
|---|---|---|

Figure 2: Comparison of the performance of **no-think** mode of Hybrid model (Qwen3-8B) and Instruct model (Qwen2.5-7B-Instruct) on MATH500, AIME24, and GPQA. We show bar plots of accuracy, average output length and wait count across these three datasets.

think mode than in the no-think mode (e.g. 63% accuracy and 11,394 tokens in think vs. 24% and 4,062 in no-think), suggesting that they do learn to use control tokens to modulate outputs.

However, comparison with Qwen2.5-7B-Instruct shows that thinking mode switch is incomplete: Qwen2.5-7B-Instruct generates short outputs without reasoning. But, in the "*no_think*" mode, Qwen3-8B still generate outputs with reasoning, which leaks reasoning words (e.g., "wait", "hmm") outside empty "`<think>`" blocks (Figure 1). Therefore, hybrid thinking affords only limited control, motivating further study of training strategies and their trade-offs.

To investigate which factors affect the controllability of hybrid thinking—i.e., the balance between think and no-think mode—we systematically analyze and obtain the following findings:

1. *Large-scale data enables effective hybrid switching.* A sufficiently large amount of hybrid thinking data (e.g., 140k samples) is required for stable control.

2. *No-think control improves with non-paired data.* Using think and no-think answers from different questions (*no-pairs*) yields stronger controllability than paired settings (think and no-think answers from the same questions).

3. *A moderate rise in no-think data strengthens control.* Appropriately increasing the no-think data proportion reduces output length in the no-think mode while maintaining accuracy.

4. *Two-phase training enhances no-think control.* First training on pure think data, then applying hybrid training, further improves no-think controllability.

Finally, we compare hybrid thinking models with those trained purely on think or no-think data, under carefully matched training settings. Our results show that hybrid thinking cannot achieve complete mode separation, as the no-think mode remains influenced by think data. Based on our

analysis of training factors, we propose a practical recipe to improve controllability in hybrid thinking, which maintains accuracy in both two modes while substantially reducing no-think verbosity. For example, on MATH500, the average no-think output length decreases from 1085 to 585 tokens, and the number of "wait" occurrences is reduced from 5917 to 522. These results demonstrate that our recipe effectively improves controllability without sacrificing performance, highlighting both the limitations of current hybrid thinking and concrete directions for its advancement.

We thus advocate that future hybrid thinking training should deliberately allocate appropriately more no-think data and adopt structured strategies such as two-phase training, which we believe will be crucial for advancing the development of controllable and efficient reasoning models.

## 2 MOTIVATIONS: NO-THINK MODE IS STILL THINKING

The implementation of hybrid thinking is typically achieved by adding control tokens (e.g., "\no_think", "\think") into prompts and applying supervised fine-tuning (SFT), enabling models to seemingly decide whether to reason. To examine whether this provides true controllability, we evaluate Qwen3-8B (Yang et al., 2025) on MATH500 (Lightman et al., 2023), AIME24, and GPQA (Rein et al., 2024), with Qwen2.5-7B-Instruct serving as a pure no-think baseline. Evaluation results in Table 1 show that hybrid models consistently perform better in the think mode than in the no-think mode (e.g., Qwen3-8B achieves 63% accuracy with an average length of 11,394 tokens in think mode, compared to 24% and 4,062 tokens in no-think mode on AIME24), confirming that they learn to use control tokens to modulate output style and length.

However, comparison with Qwen2.5-7B-Instruct reveals that mode separation remains incomplete. The Instruct model produces much shorter no-think outputs (e.g., 1,232 tokens on AIME24, compared to 4,062 tokens for Qwen3-8B in no-think mode) and does so strictly without reasoning traces. In contrast, Qwen3-8B still leaks reasoning-like behaviors in its no-think responses. Across the AIME24 no-think evaluation set, Qwen3-8B generated 646 "wait" tokens.

As further illustrated in Figure 1, when answering an AIME24 problem, Qwen3-8B, despite leaving the "<think>" reasoning block empty in no-think mode, inserted reasoning phrases such as "wait" when summarizing before the final answer. These findings indicate that hybrid models can only partially suppress reasoning and fail to achieve a clean separation between think and no-think modes. Motivated by this limitation, our work focuses on systematically exploring training strategies and trade-offs to strengthen mode controllability, aiming to achieve finer and more reliable separation between think and no-think behaviors.

## 3 WHAT FACTORS AFFECT HYBRID THINKING MODEL TRAINING?

This section investigates training factors influencing the supervised fine-tuning (SFT) of hybrid thinking. (1) *Data Scale:* effective control requires sufficiently large data (e.g., 140k samples). (2) *Pairing:* avoiding paired think/no-think samples—i.e., using answers from different questions—slightly improves no-think controllability compared to paired data. (3) *Data Ratio:* while models easily learn the think mode, no-think is harder; moderately increasing its proportion enhances controllability. (4) *Two-Phase Training:* training first on think data and then applying think-mode fusion yields stronger no-think control than mixed training..

All experiments use the OpenR1-Math (Hugging Face, 2025) dataset, with no-think data from Numina-Math (LI et al., 2024) and think data generated by DeepSeek-R1 (Guo et al., 2025).

### 3.1 LARGE-SCALE DATA FACILITATES HYBRID THINKING SWITCHING

We first explore the impact of *data scale* on training hybrid thinking models. Specifically, we collect 20k, 40k, 80k, and 140k examples, ensuring that the ratio between think and no-think data is approximately 1:1. For each question, we include both a think-mode and a no-think-mode response. We use Qwen2.5-7B-Instruct for training, and the results are presented in Table 2 and Figure 3, and for Llama3.1-8B-Instruct, the results are presented in Table 9.

We evaluate the model on MATH500, AIME24, and GPQA under both think and no-think modes. The results show that as the training data scale increases, the accuracy in the think mode steadily

Table 2: Performance of Qwen2.5-7B-Instruct with different data scales on MATH500, AIME24, and GPQA. "Instruct" refers to the original Qwen2.5-7B-Instruct model, while 20k/40k/80k/140k denote the number of training samples used. All training sets maintain a nearly 1:1 ratio of think to no-think data. We report accuracy and average output length under both modes for each model.

| Data Scale | MATH500 | | AIME24 | | GPQA | |
|---|---|---|---|---|---|---|
| | Think | No-think | Think | No-think | Think | No-think |
| Accuracy (%) | | | | | | |
| Instruct | – | 74.96 | – | 13.33 | – | 31.01 |
| 20k | 83.18 | 60.12 | 18.33 | 7.00 | 40.00 | 39.50 |
| 40k | 85.46 | 62.10 | 19.33 | 7.00 | 40.61 | 38.99 |
| 80k | 85.88 | 63.16 | 27.67 | 5.33 | 40.25 | 38.89 |
| 140k | 86.58 | 63.90 | 36.00 | 5.00 | 41.01 | 34.95 |
| Average Output Length | | | | | | |
| Instruct | – | 608.85 | – | 1272.64 | – | 9.77 |
| 20k | 4704.91 | 2214.08 | 13397.91 | 5654.04 | 9444.03 | 5503.32 |
| 40k | 4589.04 | 1437.72 | 13586.36 | 2584.13 | 9571.43 | 5330.81 |
| 80k | 4539.53 | 1086.00 | 12799.13 | 2086.05 | 9712.00 | 4811.29 |
| 140k | 4442.49 | 775.67 | 12507.64 | 1293.08 | 9826.74 | 4583.18 |

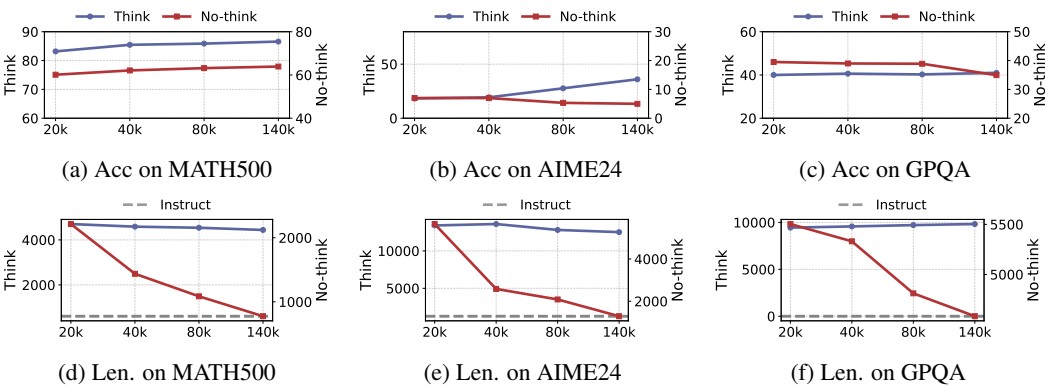

(a) Acc on MATH500     (b) Acc on AIME24     (c) Acc on GPQA

(d) Len. on MATH500     (e) Len. on AIME24     (f) Len. on GPQA

Figure 3: Line plots of Qwen2.5-7B-Instruct performance with different training data scales (20k/40k/80k/140k) on MATH500, AIME24, and GPQA. We report accuracy (top) and average output length (bottom) under both think and no-think modes. The results show that while accuracy and output length in the think mode, as well as accuracy in the no-think mode, remain relatively stable across scales, the no-think output length decreases substantially with larger training data.

improves, while the output length remains roughly unchanged. In contrast, the accuracy in the no-think mode remains relatively stable, but the output length decreases significantly. At 140k scale, the no-think output length is the shortest, approaching that of the baseline. For example, on MATH500, after training with 140k data, the model's no-think output length is reduced to 775.67, which is nearly comparable to the baseline length of 608.85. This indicates that training with 140k examples serves as an effective data scale for enabling the model to achieve hybrid thinking.

## 3.2 USING DIFFERENT QUESTIONS FOR THINK AND NO-THINK ANSWERS IMPROVES THINKING MODE CONTROL

We further investigate the impact of *With-Think vs No-Think Pairs* on hybrid thinking training. Using 20k and 40k training examples, we compare two settings: one where each question is paired with both a think-mode and a no-think-mode response, and another where the think and no-think responses are not paired to the same question.

Table 3: Comparison between pairs and no-pairs on MATH500 and AIME24 under think and no-think modes. Here, *pairs* means each question has both think and no-think responses, while *no-pairs* means the responses come from different questions. We report accuracy and average output length. Notably, the no-pairs setting consistently yields the shortest no-think outputs across scales, while maintaining comparable accuracy to pairs in both modes.

| Scale | Setting | MATH500 | | | | AIME24 | | | |
| | | Think | | No-think | | Think | | No-think | |
| | | Acc. | Len. | Acc. | Len. | Acc. | Len. | Acc. | Len. |
| 20k | pairs | 83.18 | 4704.91 | 60.12 | 2214.08 | 18.33 | 13397.91 | 7.00 | 5654.04 |
| | no-pairs | 83.44 | 4733.91 | 54.26 | **1597.64** | 23.00 | 12913.75 | 5.33 | **2560.70** |
| 40k | pairs | 85.46 | 4589.04 | 62.10 | 1437.72 | 19.33 | 13586.36 | 7.00 | 2584.13 |
| | no-pairs | 84.40 | 4560.66 | 61.76 | **942.87** | 24.00 | 13308.04 | 7.67 | **1912.82** |
| 80k | pairs | 85.88 | 4539.53 | 63.16 | 1086.00 | 27.67 | 12799.13 | 5.33 | 2086.05 |
| | no-pairs | 86.6 | 4584.13 | 56.12 | **884.79** | 32.67 | 12765.81 | 4.0 | **1146.767** |

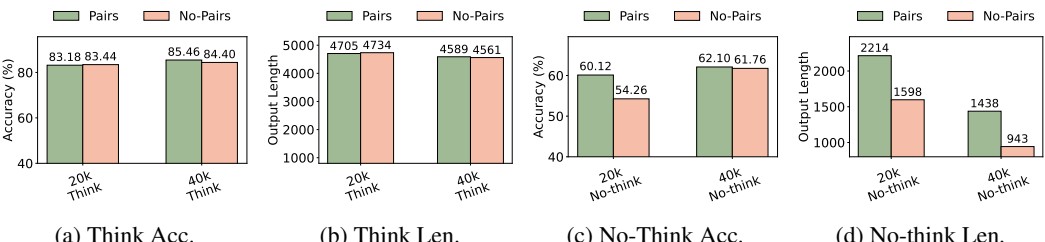

(a) Think Acc.          (b) Think Len.          (c) No-Think Acc.          (d) No-think Len.

Figure 4: Comparison of accuracy and output lengths under think and no-think modes on MATH500. The bar charts illustrate model behavior under different training settings (pairs vs. no-pairs), highlighting how no-pairs could reduce output length in no think mode.

The results, shown in Table 3 Table 10 and Figure 5, indicate that under the *no-pairs* setting, models produce substantially shorter outputs in the no-think mode, while maintaining almost the same accuracy in both think and no-think modes. For example, at the 40k scale, the think accuracy of pairs and no-pairs is 85 and 84 respectively, and the no-think accuracy is 62% and 61%. However, the no-think output length of the no-pairs setting is only 942, significantly shorter than that of the pairs setting. This suggests that when constructing datasets, it is preferable to use non-paired settings—i.e., ensuring that think and no-think responses do not originate from the same question.

### 3.3 MODERATELY HIGHER NO-THINK DATA PROPORTIONS IMPROVE NO-THINK CONTROL

From Table 2, we observe that think-mode accuracy changes little with increasing data scale, whereas the no-think output length decreases substantially as the scale grows in MATH500. This suggests a practical lever for controllability: upweighting no-think data may strengthen the model's ability to remain concise in the no-think mode, thereby improving overall hybrid-thinking control.

To validate this hypothesis, we fine-tune the Qwen2.5-7B and Llama3.1-8B model with 80k training examples while controlling the ratio between think and no-think data, and then evaluate its hybrid thinking ability on MATH500 and AIME24. The results are shown in Table 4, Table 12 and Figure 5. We observe that appropriately increasing the proportion of no-think data improves the controllability of the no-think mode, as evidenced by shorter outputs, while leaving the accuracy and output length of the think mode, as well as the accuracy of the no-think mode, largely unaffected.

### 3.4 TWO-PHASE TRAINING: TRAINING ON THINK DATA FOLLOWED BY HYBRID THINKING TRAINING ENHANCES NO-THINK CONTROL

In the previous sections, we collected both think and no-think data and directly shuffled them for fine-tuning, enabling the model to acquire hybrid thinking ability, which we called as mix training.

Table 4: Effect of different think-to-no-think data ratios on MATH500 and AIME24 under think and no-think modes. Ratios denote the proportion of think to no-think samples, with the total training size fixed at 80k. Increasing the proportion of no-think data effectively reduces no-think output length while largely preserving accuracy in both modes. *Note: The "1:1" ratio is an approximate proportion ($\approx 5\!:\!4$) due to dataset construction.*

| | MATH500 | | | | AIME24 | | | |
| | Think | | No-think | | Think | | No-think | |
| Think:No-think | Acc. | Len. | Acc. | Len. | Acc. | Len. | Acc. | Len. |
|---|---|---|---|---|---|---|---|---|
| 1:4 | 83.14 | 4843.01 | 61.16 | 788.70 | 20.56 | 13229.06 | 4.33 | **1453.21** |
| 1:2 | 84.70 | 4577.90 | 66.80 | **761.30** | 25.00 | 13233.21 | 3.00 | 1592.96 |
| 1:1 | 85.88 | 4539.53 | 63.16 | 1086.00 | 27.67 | 12799.13 | 5.33 | 2086.05 |
| 2:1 | 86.6 | 4452.42 | 62.48 | 1121.09 | 31.33 | 12524.66 | 3.33 | 2233.13 |

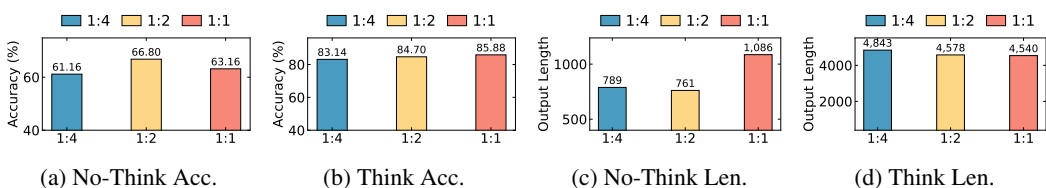

| (a) No-Think Acc. | (b) Think Acc. | (c) No-Think Len. | (d) Think Len. |

Figure 5: Comparison of output lengths under think and no-think modes on MATH500. The bar charts illustrate model behavior under different think-to-no-think data ratios, showing that increasing the proportion of no-think data reduces output length in the no-think mode.

In contrast, following the training strategy of models such as Qwen3-32B(Yang et al., 2025), hybrid thinking can also be achieved through a *two-phase training* process: first training the model on think-mode data to acquire reasoning ability, and then applying fine-tuning with both think and no-think data as a *"Thinking Mode Fusion"* phase to integrate the no-think capabilities.

To compare the effectiveness of two-phase training (think first, then think-mode fusion) against the random mixing strategy, we fine-tune the Qwen2.5-7B-Instruct and Llama3.1-8B-Instruct model with 80k examples, keeping the ratio of think and no-think pairs roughly 1:1 (an approximate proportion of 5:4). As shown in Table 5, Table 13 and Figure 6, We observe that under different data scales, two-phase training consistently reduces output length in the no-think mode compared to mix training. For example, with 20k data, the no-think output length is reduced to 870 on MATH500 and 1847 on AIME24, compared to 2214 and 5654 under mix training.

The results demonstrate that two-phase training effectively reduces the output length in the no-think mode while preserving the model's think-mode accuracy. This suggests that two-phase training mitigates the influence of think-mode data on no-think outputs, thereby improving no think control.

## 4 HOW CAN WE ACHIEVE TRUE HYBRID THINKING?

The previous sections mainly examined the training factors that influence hybrid thinking. In this section, we first compare hybrid thinking models with pure thinking and pure no-thinking models to analyze the trade-offs inherent in hybrid thinking. We also evaluate open-source models such as Qwen3 for further insights. Building on these analyses, we then propose a training recipe inspired by the findings of the previous section, with the goal of enhancing control over the no-think mode.

**Experimental Setups**. We sample 80K examples from the *OpenR1-Math* default subset to construct three datasets: Pure think (only `think` responses), Pure no-Think (only `no_think` responses), and Hybrid think (a $5\!:\!4$ mixture of `think` and `no_think`). Both *Qwen2.5-7B-Base* and *LLaMA-3.1-8B-Base* are fine-tuned on each dataset for 3 epochs with a learning rate of $1.0 \times 10^{-5}$ and a warm-up ratio of $0.1$, while all other optimization settings remain at their defaults.

Table 5: Comparison of 2-phase and mix training strategies on MATH500 and AIME24 under think and no-think modes. Here, *Scale* denotes the number of training samples (e.g., 20k means 20,000 samples). We report both accuracy and average output length. The *2-phase* strategy first trains on pure-think data before fine-tuning with mixed think and no-think data, while the *mix* strategy directly trains on think and no-think mixed data. Notably, across all scales, the *2-phase* strategy consistently yields shorter output lengths under the no-think mode compared to *mix* training.

| Scale | Setting | MATH500 | | | | AIME24 | | | |
| | | Think | | No-think | | Think | | No-think | |
| | | Acc. | Len. | Acc. | Len. | Acc. | Len. | Acc. | Len. |
| 20k | 2-phase | 83.38 | 4797.40 | 47.26 | **870.50** | 20.00 | 13620.71 | 4.33 | **1847.35** |
| | mix | 83.18 | 4704.91 | 60.12 | 2214.08 | 18.33 | 13397.91 | 7.00 | 5654.04 |
| 40k | 2-phase | 85.18 | 4670.88 | 43.52 | **798.87** | 22.67 | 13562.19 | 1.67 | **1635.34** |
| | mix | 85.46 | 4589.04 | 62.10 | 1437.72 | 19.33 | 13586.36 | 7.00 | 2584.13 |
| 80k | 2-phase | 85.86 | 4626.42 | 51.86 | **639.60** | 30.00 | 12962.41 | 3.00 | **1398.48** |
| | mix | 85.88 | 4539.53 | 63.16 | 1086.00 | 27.67 | 12799.13 | 5.33 | 2086.05 |
| 140k | 2-phase | 86.78 | 4481.74 | 63.60 | **585.63** | 32.67 | 12271.01 | 5.00 | **853.82** |
| | mix | 86.58 | 4442.49 | 63.90 | 775.67 | 36.00 | 12507.64 | 5.00 | 1293.08 |

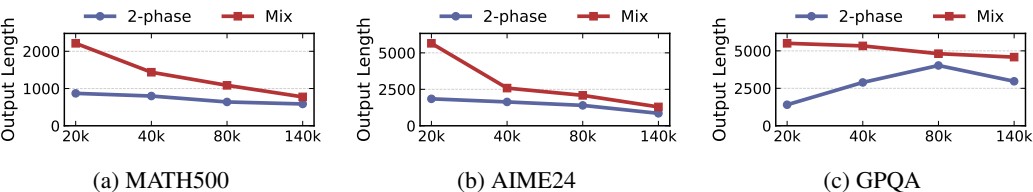

(a) MATH500 (b) AIME24 (c) GPQA

Figure 6: Average output length of models trained with 2-phase and mix strategies across three benchmarks (MATH500, AIME24, and GPQA) under different data scales with no think mode.

### 4.1 LIMITATIONS OF CURRENT HYBRID THINKING TRAINING ON MODE SWITCHING

Since current hybrid thinking models lack direct pure-think or pure no-think counterparts, we train three variants from the same base model: a hybrid model (80k hybrid data), a pure-think model (80k think data), and a pure no-think model (80k no-think data). As shown in Table 6, the hybrid model achieves nearly identical performance to the pure-think model in the think mode (e.g., MATH500 output length 4539 vs. 4340; AIME24 output length 12324 vs. 12324), suggesting that think-mode ability is unaffected. However, in the no-think mode, hybrid models achieve higher accuracy but generate substantially longer outputs than pure no-think models, indicating interference from think data that undermines controllability.

We further examine open-source hybrid models such as Qwen3 in Table 7. Even with large-scale training data, these models still exhibit reasoning-related behaviors in the no-think mode. To quantify this phenomenon, we count the occurrences of reasoning-supportive tokens (e.g., "wait", "hmm", "alternatively") and observe frequent appearances across no-think outputs. For instance, all Qwen3 models (4B, 8B, and 14B) produce such tokens in the no-think setting.

These findings underscore a fundamental limitation of current hybrid thinking models: they only partially suppress reasoning and thus fail to guarantee clean separation between modes.

### 4.2 NEW RECIPE FOR IMPROVING NO THINK CONTROL BASED ON OUR FINDINGS

Although hybrid thinking models cannot yet achieve 100% controllability over whether to engage in reasoning, the analysis in previous sections suggests practical strategies to improve their performance. The results on MATH500 and MMLU-STEM are reported in Table 8.

Table 6: Comparison of hybrid, pure-think, and pure-no-think models on MATH500 and AIME24 under think and no-think modes. We report accuracy (Acc.), output length (Len.), and #Wait count (i.e., occurrences of reflection words such as "wait", "hmm", "alternatively"). Results show that in the no-think mode, hybrid models still produce much longer outputs than pure no-think models and generate frequent reflection words.

| Family | Model Type | Think | | | No-think | | |
|---|---|---|---|---|---|---|---|
| | | Acc. | Len. | #Wait | Acc. | Len. | #Wait |
| | | MATH500 | | | | | |
| Qwen2.5-7B | Pure think | 86.66 | 4340.62 | 99149 | – | – | – |
| | Pure no-think | – | – | – | 53.10 | 430.44 | 0 |
| | Hybrid think | 86.12 | 4539.38 | 105655 | 77.42 | 2971.03 | **64622** |
| LLaMA3.1-8B | Pure think | 85.52 | 4079.28 | 127763 | – | – | – |
| | Pure no-think | – | – | – | 21.00 | 509.92 | 0 |
| | Hybrid think | 72.18 | 6152.42 | 196369 | 42.76 | 769.91 | **1093** |
| Phi-14B | Pure think | 91.96 | 3291.77 | 7717 | – | – | – |
| | Pure no-think | – | – | – | 80.08 | 719.81 | 1 |
| | Hybrid think | 92.34 | 3534.30 | 110821 | 67.92 | 515.12 | **11** |
| | | AIME24 | | | | | |
| Qwen2.5-7B | Pure think | 33.33 | 12324.62 | 14460 | – | – | – |
| | Pure no-think | – | – | – | 2.33 | 583.10 | 0 |
| | Hybrid think | 29.33 | 12615.44 | 15217 | 5 | 853.82 | **1** |
| LLaMA3.1-8B | Pure think | 40.67 | 11786.05 | 30064 | – | – | – |
| | Pure no-think | – | – | – | 1.00 | 631.82 | 0 |
| | Hybrid think | 10.00 | 13850.14 | 23902 | 3.00 | 1288.76 | **96** |
| Phi-14B | Pure think | 73.33 | 7136.53 | 1776 | – | – | – |
| | Pure no-think | – | – | – | 20.00 | 1653.98 | 3 |
| | Hybrid think | 56.67 | 10148.61 | 18020 | 10.33 | 884.72 | **0** |

Table 7: Performance of Qwen3 models on MATH500 and AIME24 under think and no-think modes. We report accuracy (Acc.), output length (Len.), and #Wait count. Results show that even open-source models still produce such reflection words in the no-think mode.

| Model | Mode | MATH500 | | | AIME24 | | |
|---|---|---|---|---|---|---|---|
| | | Acc. | Len. | #Wait | Acc. | Len. | #Wait |
| Qwen3-8B | Think | 92.82 | 4384.35 | 83703 | 63.33 | 11394.54 | 12184 |
| | No-think | 82.90 | 958.31 | **646** | 24.00 | 4061.67 | **184** |
| Qwen3-4B | Think | 92.02 | 4679.54 | 97998 | 61.67 | 11595.51 | 13754 |
| | No-think | 82.22 | 1004.13 | **611** | 20.67 | 4636.37 | **762** |
| Qwen3-14B | Think | 93.96 | 4450.98 | 65320 | 29.33 | 3497.86 | 11091 |
| | No-think | 85.88 | 886.17 | **498** | 67.67 | 11401.06 | **295** |

Here we use the pure-think model and the Qwen2.5-7B-Instruct model as baselines, representing the best achievable performance in the think and no-think modes, respectively. Inspired by the findings in Section 3, we take the 80k mixed training as the original recipe and compare it with our new 2-phase recipe trained on 140k data with pairs. Here, our recipe adopts paired data as a trade-off: as discussed in Section 3.2, using unpaired think/no-think data can slightly improve controllability under the same data scale, but each instance is then used only once—either for think or for no-think—thus limiting the overall dataset size. We therefore prioritize enlarging the dataset at the cost of pairing, and the final results show that this scale-oriented recipe yields better overall performance. As shown, our recipe achieves nearly identical accuracy and output length to the original recipe in the think mode, but in the no-think mode it substantially reduces verbosity while

Table 8: Performance of different training settings on MATH500 and MMLU-STEM under think and no-think modes. We report accuracy (Acc.), output length (Len.), and #Wait count (i.e., occurrences of reflection words such as "wait", "hmm", "alternatively"). Our recipe maintains accuracy while substantially reducing no-think verbosity and reflective tokens.

| | MATH500 | | | | | |
|---|---|---|---|---|---|---|
| Model | Think | | | No-think | | |
| | Acc. | Len. | #Wait | Acc. | Len. | #Wait |
| pure-think | 86.66 | 4340.62 | 99149 | – | – | – |
| Instruct | – | – | – | 75.42 | 640.91 | 0 |
| Original Recipe | 85.88 | 4539.53 | 116663 | 63.16 | 1085.99 | 5917 |
| Our Recipe | 86.78 | 4481.74 | 103335 | 63.60 | **585.63** | **522** |

| | MMLU-STEM | | | | | |
|---|---|---|---|---|---|---|
| Model | Think | | | No-think | | |
| | Acc. | Len. | #Wait | Acc. | Len. | #Wait |
| pure-think | 84.97 | 2924.20 | 47382 | – | – | – |
| Instruct | – | – | – | 66.25 | 427.01 | 18 |
| Original Recipe | 84.68 | 3129.53 | 52665 | 80.97 | 2014.49 | 24653 |
| Our Recipe | 84.74 | 3141.14 | 50574 | 60.61 | **956.57** | **4017** |

maintaining accuracy—for example, on MATH500 the average output length decreases from 1085 to 585 tokens, and the number of "wait" occurrences drops from 5917 to 522.

## 5 RELATED WORKS

**LLM Reasoning.** Current approaches to enhancing reasoning in language models (Chen et al., 2025a; Plaat et al., 2024; Sun et al., 2023) mainly fall into reinforcement learning (RL) (Schulman et al., 2017) and supervised fine-tuning (SFT) (Jaech et al., 2024; Yang et al., 2024). In RL, DeepSeek (Guo et al., 2025; Liu et al., 2024) achieved state-of-the-art performance with GRPO (Shao et al., 2024; Yu et al., 2025) and distilled reasoning traces to smaller models, inspiring replications such as Logic RL (Xie et al., 2025) and SimpleRL-Zoo (Zeng et al., 2025). Techniques to further enhance reasoning during RL are also explored (Baek & Tegmark, 2025; Yeo et al., 2025). In SFT, curated reasoning traces are widely used, e.g., SkyThought-T1 (Team, 2025b) and Bespoke-Stratos-32B (Labs, 2025). Analyses show that structure of reasoning steps may matter more than content (Li et al., 2025a), while initial tokens play a key role (Ji et al., 2025). Data selection is also critical, with s1 (Muennighoff et al., 2025) showing that small high-quality sets yield large gains. Other works examine additional SFT factors (Chen et al., 2025b; 2024a; Tian et al., 2025; Liu et al., 2025). Overall, RL and SFT provide complementary pathways to improve reasoning, highlighting the importance of training objectives, data structure, and sample quality.

**Efficient Reasoning.** Current reasoning models still face efficiency challenges, with many producing excessively long outputs (Bandyopadhyay et al., 2025; Li et al., 2025b). Early efforts such as Kimi 1.5 (Team et al., 2025b) and Sky-Thought (Team, 2025a) reduce verbosity by aligning long and short responses via preference optimization, while methods like TokenSkip (Xia et al., 2025) and LightThinker (Zhang et al., 2025) improve efficiency by removing redundant tokens or compressing intermediate thoughts. Beyond shortening reasoning, hybrid thinking (Sui et al., 2025; Chen et al., 2024b) aims to control *when* to reason by introducing control tokens (e.g., \think, \nothink), a strategy now adopted by models such as Gemini, Qwen3, and DeepSeek V3.1 . However, current implementations still show imperfect mode separation, often leaking reasoning behavior into no-think outputs. Other works explore efficiency from different angles, such as early stopping with probing (Fu et al., 2024) or analyzing overthinking behaviors (Wang et al., 2025; Sui et al., 2025).

**Hybrid Thinking.** Hybrid thinking has recently emerged as a mechanism to balance efficiency and reasoning. It was first popularized by Gemini (Team et al., 2025a) through control tokens that switch between direct answering and reasoning, and later extended by Qwen3 (Yang et al., 2025)

across multiple model scales. GPT-oss (Agarwal et al., 2025) further explored varying degrees of reasoning depth, while DeepSeek V3.1 (Liu et al., 2024) advanced the paradigm with large-scale reinforcement learning for stronger controllability and efficiency. Despite these advances, little work has systematically examined the training factors and trade-offs underlying hybrid thinking.

## 6 CONCLUSION

In this paper, we systematically analyzed hybrid thinking models, which balance efficiency and reasoning by switching between think and no-think modes. Our evaluation shows that while hybrid thinking enables partial controllability, current models still fail to fully separate the two modes. Through controlled experiments, we identified four key factors—data scale, paired vs. unpaired answers, data ratio, and two-phase training—with larger no-think proportions and two-phase training proving most effective. We conclude that hybrid thinking entails inherent trade-offs compared to pure-think or pure-no-think models, and recommend allocating moderately more no-think data and refining training strategies to improve controllability.

**Limitations**. Our experiments primarily focus on the Qwen2.5-7B model, which may limit the generality of our findings. Future work could extend the analysis to larger-scale models and a broader range of architectures to further validate and refine the conclusions.

ETHICS STATEMENT

This work uses only publicly available datasets (MATH500, AIME24, GPQA, MMLU-STEM) under their licenses and does not involve human subjects or private data. Our analysis focuses on controllability of reasoning behaviors in LLMs. We acknowledge potential risks of misuse (e.g., generating misleading reasoning), but our findings aim to improve safety and efficiency in model design.

REPRODUCIBILITY STATEMENT

All datasets are standard public benchmarks. Experimental settings, hyperparameters, and ablation studies are detailed in the main text and Appendix. Code and processed data will be released with the camera-ready version to ensure full reproducibility.

LLM USAGE STATEMENT

LLMs were used only for proofreading, editing LaTeX, and improving readability. All technical ideas, experiments, and conclusions are the authors' own. The authors take full responsibility for the final content.

ACKNOWLEDGMENTS

This research was supported in part by NSF awards 2112606 and 2117439

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

# APPENDIX

## ADDITIONAL RESULTS ON FACTORS AFFECT HYBRID THINKING

Table 9 reports full results of **Llama3.1-8B-Instruct** trained using both two-phase and naive-mix training across scales on Math500, AIME24, and GPQA-Diamond.

Table 9: Performance of Llama3.1-8B-Instruct with different data scales on MATH500, AIME24, GPQA, and MMLU-STEM. "Instruct" refers to the original Llama3.1-8B-Instruct model, while 20k/40k/80k/140k denote the number of training samples used. All training sets maintain a nearly 1:1 ratio (an approximate proportion of 5:4) of think to no-think data. We report accuracy and average output length under both modes for each model.

| Data Scale | MATH500 | | AIME24 | | GPQA | | MMLU | |
|---|---|---|---|---|---|---|---|---|
| | Think | No-think | Think | No-think | Think | No-think | Think | No-think |
| | Accuracy (%) | | | | | | | |
| Instruct | – | 49.32 | – | 6.0 | – | 29.29 | – | 52.11 |
| 20k | 69.10 | 49.12 | 8.33 | 5.67 | 36.41 | 32.53 | 76.82 | 59.88 |
| 40k | 73.40 | 53.92 | 10.00 | 4.00 | 36.52 | 30.35 | 78.50 | 55.22 |
| 80k | 78.12 | 55.22 | 20.33 | 4.00 | 37.58 | 31.16 | 79.16 | 48.84 |
| 140k | 82.00 | 57.72 | 23.67 | 2.67 | 34.95 | 31.67 | 78.43 | 52.52 |
| | Average Output Length | | | | | | | |
| Instruct | – | 2426.36 | – | 7153.73 | – | 3190.01 | – | 911.70 |
| 20k | 6008.2 | 1111.1 | 14182.8 | 2949.1 | 9119.5 | 2814.4 | 3632.3 | 1127.2 |
| 40k | 5685.3 | 1118.5 | 14088.2 | 2995.5 | 10144.5 | 1957.3 | 3866.0 | 765.6 |
| 80k | 5263.7 | 899.7 | 13253.2 | 2108.0 | 10004.7 | 987.2 | 3775.0 | 546.1 |
| 140k | 4896.8 | 780.9 | 13141.8 | 2229.5 | 10190.1 | 3352.6 | 3782.8 | 849.4 |

Table 10 reports full results of **Llama3.1-8B-Instruct** trained using paired dataset and unpaired dataset across scales on Math500, AIME24.

Table 10: Comparison between pairs and no-pairs on MATH500 and AIME24 under think and no-think modes. Here, *pairs* means each question has both think and no-think responses, while *no-pairs* means the responses come from different questions. We report accuracy and average output length. Notably, the no-pairs setting consistently yields the shortest no-think outputs across scales, while maintaining comparable accuracy to pairs in both modes.

| Scale | Setting | MATH500 | | | | AIME24 | | | |
|---|---|---|---|---|---|---|---|---|---|
| | | Think | | No-think | | Think | | No-think | |
| | | Acc. | Len. | Acc. | Len. | Acc. | Len. | Acc. | Len. |
| 20k | pairs | 69.10 | 6008.17 | 49.12 | **1111.11** | 8.33 | 14182.81 | 5.67 | 2949.06 |
| | no-pairs | 69.50 | 6187.14 | 51.70 | 1159.26 | 11.33 | 13880.53 | 3.33 | **2850.43** |
| 40k | pairs | 73.40 | 5685.25 | 53.92 | 1118.49 | 10.00 | 14088.24 | 4.00 | **2995.47** |
| | no-pairs | 73.52 | 5679.92 | 54.34 | **1096.61** | 13.33 | 13926.53 | 3.00 | 3203.56 |
| 80k | pairs | 78.12 | 5263.71 | 55.22 | 899.72 | 20.33 | 13253.21 | 4.00 | 2108.03 |
| | no-pairs | 77.88 | 5353.67 | 56.32 | **864.91** | 18.00 | 13437.94 | 2.67 | **1877.41** |

Table 11 reports full results of **Llama3.1-8B-Instruct** trained using paired dataset and unpaired dataset across scales on Math500, AIME24.

Table 11 reports full results of **Llama3.1-8B-Instruct** trained using paired dataset and unpaired dataset across scales on GPQA, MMLU.

Table 13 reports full results of **Llama3.1-8B-Instruct** trained using 2-phase training and mix training strategies across scales on Math500, AIME24.

Table 11: Effect of different think-to-no-think data ratios on MATH500 and AIME24 under think and no-think modes. Ratios denote the proportion of think to no-think samples, with the total training size fixed at 80k. Increasing the proportion of no-think data effectively reduces no-think output length while largely preserving accuracy in both modes. *Note: The "1:1" ratio is an approximate proportion ($\approx 5{:}4$) due to dataset construction.*

| | MATH500 | | | | AIME24 | | | |
| | Think | | No-think | | Think | | No-think | |
| Think:No-think | Acc. | Len. | Acc. | Len. | Acc. | Len. | Acc. | Len. |
|---|---|---|---|---|---|---|---|---|
| 1:4 | 69.56 | 6184.32 | 50.40 | **802.80** | 5.33 | 14416.15 | 2.00 | **1234.35** |
| 1:2 | 74.32 | 5743.64 | 53.04 | 811.43 | 13.33 | 13767.91 | 3.33 | 1774.01 |
| 1:1 | 78.12 | 5263.71 | 55.22 | 899.72 | 20.33 | 13253.21 | 4.00 | 2108.03 |
| 2:1 | 77.96 | 5147.29 | 55.16 | 1025.80 | 19.33 | 13328.44 | 4.67 | 2229.22 |

Table 12: Effect of different think-to-no-think data ratios on GPQA and MMLU under think and no-think modes. Ratios denote the proportion of think to no-think samples, with the total training size fixed at 80k. Increasing the proportion of no-think data effectively reduces no-think output length while largely preserving accuracy in both modes. *Note: The "1:1" ratio is an approximate proportion ($\approx 5{:}4$) due to dataset construction.*

| | GPQA | | | | MMLU | | | |
| | Think | | No-think | | Think | | No-think | |
| Think:No-think | Acc. | Len. | Acc. | Len. | Acc. | Len. | Acc. | Len. |
|---|---|---|---|---|---|---|---|---|
| 1:4 | 32.73 | 10886.36 | 30.81 | 1470.06 | 75.45 | 4474.87 | 50.02 | 568.52 |
| 1:2 | 33.23 | 10493.00 | 31.11 | 1039.15 | 77.74 | 4181.22 | 49.60 | **540.94** |
| 1:1 | 37.58 | 10004.75 | 31.16 | **987.24** | 79.16 | 3774.97 | 48.84 | 546.10 |
| 2:1 | 38.48 | 9885.08 | 31.92 | 1359.05 | 79.80 | 3542.41 | 49.67 | 584.47 |

Table 14 reports full results of **Llama3.1-8B-Instruct** trained using 2-phase training and mix training strategies across scales on GPQA and MMLU-STEM.

ADDITIONAL RESULTS ON TRAINING STRATEGIES

Table 15 reports full results of two-phase and naive-mix training across scales on Math500, AIME24, and GPQA-Diamond. Two-phase consistently reduces no-think output length while preserving think-mode accuracy, showing its advantage in suppressing unnecessary reasoning traces without harming task performance.

SYSTEM PROMPT ABLATION

Table 16 compares Qwen 2-phase training with and without system prompts on Math500, AIME24, and GPQA-Diamond. Results show that while system prompts slightly reduce output length, they also lead to degraded accuracy in no-think mode.

Table 13: Comparison of 2-phase and mix training strategies on MATH500 and AIME24 under think and no-think modes. Here, *Scale* denotes the number of training samples (e.g., 20k means 20,000 samples). We report both accuracy and average output length. The *2-phase* strategy first trains on pure-think data before fine-tuning with mixed think and no-think data, while the *mix* strategy directly trains on think and no-think mixed data. Notably, across all scales, the *2-phase* strategy consistently yields shorter output lengths under the no-think mode compared to *mix* training.

| Scale | Setting | MATH500 | | | | AIME24 | | | |
| | | Think | | No-think | | Think | | No-think | |
| | | Acc. | Len. | Acc. | Len. | Acc. | Len. | Acc. | Len. |
| 20k | 2-phase | 68.56 | 6300.03 | 47.18 | 1180.40 | 7.00 | 13737.77 | 4.33 | **2735.59** |
| | mix | 69.10 | 6008.17 | 49.12 | **1111.11** | 8.33 | 14182.81 | 5.67 | 2949.06 |
| 40k | 2-phase | 72.52 | 6033.99 | 48.50 | 1170.26 | 9.67 | 14034.31 | 4.33 | **2692.68** |
| | mix | 73.40 | 5685.25 | 53.92 | **1118.49** | 10.00 | 14088.24 | 4.00 | 2995.47 |
| 80k | 2-phase | 76.90 | 5475.04 | 50.00 | **870.21** | 16.00 | 13690.04 | 5.00 | **1776.30** |
| | mix | 78.12 | 5263.71 | 55.22 | 899.72 | 20.33 | 13253.21 | 4.00 | 2108.03 |
| 140k | 2-phase | 80.66 | 5059.14 | 50.76 | **731.98** | 20.33 | 12837.85 | 1.67 | **1429.10** |
| | mix | 82.00 | 4896.76 | 57.72 | 780.87 | 23.67 | 13141.84 | 2.67 | 2229.50 |

Table 14: Comparison of 2-phase and mix training strategies on GPQA and MMLU under think and no-think modes. Here, *Scale* denotes the number of training samples (e.g., 20k means 20,000 samples). We report both accuracy and average output length. The *2-phase* strategy first trains on pure-think data before fine-tuning with mixed think and no-think data, while the *mix* strategy directly trains on think and no-think mixed data. Notably, across all scales, the *2-phase* strategy consistently yields shorter output lengths under the no-think mode compared to *mix* training.

| Scale | Setting | GPQA | | | | MMLU | | | |
| | | Think | | No-think | | Think | | No-think | |
| | | Acc. | Len. | Acc. | Len. | Acc. | Len. | Acc. | Len. |
| 20k | 2-phase | 36.31 | 10182.25 | 26.67 | **966.86** | 77.26 | 4056.91 | 53.38 | **572.71** |
| | mix | 36.41 | 9119.54 | 32.53 | 2814.42 | 76.82 | 3632.27 | 59.88 | 1127.23 |
| 40k | 2-phase | 35.15 | 10626.91 | 29.60 | **975.13** | 79.10 | 3973.26 | 53.98 | **592.56** |
| | mix | 36.52 | 10144.51 | 30.35 | 1957.28 | 78.50 | 3866.03 | 55.22 | 765.63 |
| 80k | 2-phase | 37.22 | 10100.31 | 27.98 | **705.01** | 79.10 | 3980.86 | 46.02 | **448.93** |
| | mix | 37.58 | 10004.75 | 31.16 | 987.24 | 79.16 | 3774.97 | 48.84 | 546.10 |
| 140k | 2-phase | 35.10 | 10077.06 | 28.48 | **745.44** | 78.50 | 3822.84 | 48.40 | **461.48** |
| | mix | 34.95 | 10190.07 | 31.67 | 3352.59 | 78.43 | 3782.77 | 52.52 | 849.36 |

Table 15: Performance of Qwen2.5-7B-Instruct under different training settings on Math500, AIME24, and GPQA-Diamond. We report accuracy (Acc.) and output length (Len.) for both think and no-think modes.

| Dataset | Setting | Think | | No-think | |
|---|---|---|---|---|---|
| | | Acc. | Len. | Acc. | Len. |
| Math500 | Baseline | – | – | 74.96 | 608.85 |
| | Demo (w/o pairs) | 37.08 | 650.35 | 31.68 | 390.63 |
| | 20k-2phase | 83.38 | 4797.40 | 47.26 | 870.50 |
| | 20k-mix | 83.18 | 4704.91 | 60.12 | 2214.08 |
| | 40k-2phase | 85.18 | 4670.88 | 43.52 | 798.87 |
| | 40k-mix | 85.46 | 4589.04 | 62.10 | 1437.72 |
| AIME24 | 20k-2phase | 20.00 | 13620.71 | 4.33 | 1847.35 |
| | 20k-mix | 18.33 | 13397.91 | 7.00 | 5654.04 |
| | 40k-2phase | 22.67 | 13562.19 | 1.67 | 1635.34 |
| | 40k-mix | 19.33 | 13586.36 | 7.00 | 2584.13 |
| GPQA-Diamond | 20k-2phase | 41.16 | 9872.89 | 26.72 | 1404.46 |
| | 20k-mix | 40.00 | 9444.03 | 39.50 | 5503.32 |
| | 40k-2phase | 41.62 | 9903.88 | 32.12 | 2892.41 |
| | 40k-mix | 40.61 | 9571.43 | 38.99 | 5330.81 |

Table 16: Comparison of Qwen 2-phase training with and without system prompt on Math500, AIME24, and GPQA-Diamond.

| Dataset | Model | Think | | No-think | |
|---|---|---|---|---|---|
| | | Acc. | Len. | Acc. | Len. |
| Math500 | Qwen-2-phase | 86.1 | 4539 | 77.4 | 2971 |
| | Qwen-2-phase-system-prompt | 84.6 | 4383 | 76.6 | 2840 |
| AIME24 | Qwen-2-phase | 29.3 | 12615 | 23.3 | 5744 |
| | Qwen-2-phase-system-prompt | 28.7 | 11799 | 16.0 | 6269 |
| GPQA-Diamond | Qwen-2-phase | 40.6 | 10283 | 35.9 | 8378 |
| | Qwen-2-phase-system-prompt | 39.0 | 9931 | 37.6 | 7784 |

