# OpenReview forum: "Demystifying Hybrid Thinking: Can LLMs Truly Switch Between Think and No-Think?"
_ICLR.cc/2026/Conference — ICLR 2026 Conference Withdrawn Submission_

### Official Review · Reviewer_PBRL · 2025-10-22

**Soundness:** 2
**Presentation:** 2
**Contribution:** 2
**Rating:** 2
**Confidence:** 4

**Summary:**

This paper proposes a practical recipe that, compared to standard training, can maintain accuracy in both modes while significantly reducing no-think output length (from 1085 to 585 on MATH500) and occurrences of reasoning-supportive tokens such as “wait” (from 5917 to 522 on MATH500).

**Strengths:**

Many papers analyze how LRMs can shorten "think" processes or compare the switching between "think" and "no-think" modes. To some extent, this paper fills the gap in exploring how to further reduce the length in "no-think" scenarios. Although considering words like "wait" is a very common and trivial approach—with numerous existing works already addressing this—further reducing the length in "no-think" scenarios is practically necessary.

**Weaknesses:**

This paper seems too simple. Only two simple mathematical benchmarks are considered. The authors should add more complicated math tests and coding benchmarking tests, to show the generalization of this method on no-think mode.

**Questions:**

Generalization of this method on no-think mode.

---

> ### Author Response · Authors · 2025-12-03
>
> We thank the reviewer for the comments. Below we address the main concern and the follow-up question.
>
> ---
>
> **(1) On benchmark choice and “only two simple mathematical benchmarks”.** (Weakness)
>
> We would like to clarify that our study uses **four** reasoning benchmarks rather than only two:
> **MMLU-STEM** and **MATH500** (moderate difficulty), and **AIME24** and **GPQA-Diamond** (substantially more challenging and widely regarded as hard math / science benchmarks in the literature). Thus, our evaluation already spans a range from easier STEM-style questions to highly non-trivial competition-level problems.
>
> ---
>
> **(2) On “generalization of this method on no-think mode”.** (Question)
>
> The statement *“Generalization of this method on no-think mode.”* is too vague for us to identify a specific missing experiment (e.g., whether it refers to tasks, training strategies, or architectures). Nevertheless, we have strengthened the evidence for generalization in the revised version:
>
> - In **Section 3**, we now include detailed results for **LLaMA-3.1-8B-Instruct**.
> - In **Section 4** and the Appendix, we add experiments on a larger model, **Phi-4-14B**.
>
> Across Qwen2.5-7B-Instruct, LLaMA-3.1-8B-Instruct, and Phi-4-14B, we observe the same qualitative behavior in the no-think mode and consistent gains from our training recipe, which we believe sufficiently demonstrates the method’s generalizability in the no-think setting.
>
> ---
>
> We have clarified the full set of benchmarks used and made the additional cross-model experiments explicit in the revised manuscript.

---

### Official Review · Reviewer_kdYd · 2025-10-28

**Soundness:** 3
**Presentation:** 4
**Contribution:** 2
**Rating:** 4
**Confidence:** 4

**Summary:**

The paper investigates the phenomenon that large reasoning models (LRMs) continue to produce reasoning-related tokens even in non-thinking modes.

The central insight is that hybrid thinking cannot fully disentangle reasoning and non-reasoning behaviors, as the no-think mode remains influenced by think-mode data.

Based on an analysis of contributing training factors, the authors propose a practical approach to improve controllability in hybrid thinking systems. Experimental results suggest that future hybrid thinking training should deliberately allocate a larger proportion of no-think data and adopt structured strategies such as two-phase training to achieve clearer mode separation and enhanced reasoning control.

**Strengths:**

1. The paper provides a novel perspective on hybrid reasoning by identifying that large language models tend to generate reasoning-related tokens even in non-thinking modes, offering new insights into the behavioral dynamics of reasoning control.
2. The proposed mitigation strategies—allocating a larger proportion of no-thinking data and adopting a two-phase training scheme—present practical and conceptually sound approaches to improving controllability in large reasoning model training.

**Weaknesses:**

1. The experimental evaluation lacks diversity in model size and architecture. The study is conducted primarily on Qwen2.5-7B and LLaMA3.1-8B, both relatively small and dense models, which limits the generalizability of the findings to larger or sparse (e.g., MoE) architectures.
2. The selected datasets—MATH500, AIME24, MMLU-STEM, and GPQA-Diamond—are all domain-specific and heavily focused on mathematical or scientific reasoning. The absence of simpler or general-purpose datasets, such as those involving everyday dialogue or commonsense reasoning, restricts the scope of the conclusions. It remains unclear whether reasoning leakage in the non-thinking mode is inherent to the model or primarily induced by the high difficulty level of the chosen tasks.

**Questions:**

1. The experiments are conducted exclusively on models in the 7B–8B scale. To what extent might the observed phenomena depend on model capacity or learning ability? Would larger models exhibit similar or different patterns of reasoning leakage?
2. Could the findings be extended to Mixture-of-Experts (MoE) architectures? If so, might the modular structure of MoE models help mitigate or amplify the observed reasoning leakage?
3. The selected datasets—MATH500, AIME24, MMLU-STEM, and GPQA-Diamond—are all math- and science-oriented. Would the same phenomenon persist on more general-purpose and less complex datasets (e.g., TruthfulQA)? How consistent is the reasoning leakage behavior in non-thinking modes across tasks of varying difficulty?

---

> ### Author Response · Authors · 2025-12-03
>
> We thank the reviewer for the constructive and insightful comments. Below we address each weakness and question.
>
> ---
>
> **(1) On model size, architecture diversity, and MoE vs. dense backbones. (Weakness 1)**
>
> We agree that broader model coverage is important. In the revised version, we have added experiments with **Phi-4-14B** in Section 4. Phi-4-14B is larger and architecturally different from Qwen2.5-7B-Instruct and LLaMA-3.1-8B-Instruct, yet it exhibits the same qualitative phenomena and benefits from the same training recipe, supporting the generality of our findings.
>
> Regarding MoE, recent work (e.g., *Can Mixture-of-Experts Surpass Dense LLMs Under Strictly Equal Resources?*) suggests that, under comparable training paradigms and resource budgets, MoE and dense models show similar downstream behavior. Given this, and the substantially higher cost and instability of MoE SFT, we chose to focus this study on widely used dense Models.
>
> ---
>
> **(2) On dataset choice, task difficulty, and the scope of the conclusions. (Weakness 2)**
>
> We understand the concern about using math- and science-oriented benchmarks. Our core topic, however, is **reducing reasoning leakage** in the settings where hybrid-thinking LRMs are actually deployed: challenging mathematical and scientific reasoning tasks.
>
> Most importantly, the goal of this paper is **not** to analyze or characterize when leakage first appears across all possible tasks, but to **control and mitigate** leakage where strong reasoning is required via different training factors.
>
> Within our current benchmarks we already cover a range of difficulty: MMLU-STEM and MATH500 are moderate, while AIME24 and GPQA-Diamond are substantially harder. Across these tasks, we consistently observe leakage in the no-think mode and consistent gains from our data and training choices, which is precisely the scope of our conclusions.
>
> ---
>
> **(3) Question 1: Dependence on model capacity (7B–8B vs. larger models).**
>
> As described in the response to Weakness 1, we added **Phi-4-14B** experiments in Section 4.
>
> ---
>
> **(4) Question 2: Extension to Mixture-of-Experts architectures.**
>
> As described in the response to Weakness 1, existing evidence indicates similar behavior between MoE and dense models under equal resources, and we therefore focus this work on dense LRMs and treat a systematic MoE study as future work.
>
> ---
>
> **(5) Question 3: General-purpose, less complex datasets and task difficulty.**
>
> As described in the response to Weakness 2, our study is intentionally restricted to reasoning-centric tasks of varying difficulty, where hybrid thinking is most relevant.
>
> ---
>
> We again thank the reviewer for the helpful suggestions. We have incorporated additional Phi-4-14B experiments and clarifications into the revised manuscript.

---

### Official Review · Reviewer_xR3d · 2025-10-31

**Soundness:** 2
**Presentation:** 2
**Contribution:** 2
**Rating:** 2
**Confidence:** 5

**Summary:**

This paper investigates the hybrid thinking of LLMs from the perspectives of thinking controllability, post-training and data mixing strategies. At first, the authors highlight the fact that the thinking behaviors also happen a lot when the think mode is switched off by the designated prompt. Then, the impacts of data scale and mixing mechanism are examined on the basis of Qwen models. After that, the authors propose a two-phase training strategy that first trains reasoning ability and then switch into the hybrid think training. The results show that the proposed training strategy enhance the thinking control.

**Strengths:**

1. The paper is well written and easy to follow.
2. The topic of hybrid thinking control is interesting and has a solid impact towards the development of LLMs.
3. Their results well provide with some interesting observations. The proposed two-phase training strategy show good thinking control in terms of output length.

**Weaknesses:**

1. The authors only choose Qwen2.5-7B to evaluate the impact of different factors. It is unknown whether these findings hold consistently across different models sizes and manufacturers.
2. Another interesting concern is that the output lenght is negatively correlated with the inference accuracy under the non-think mode. That is to say, the proposed training strategy is not that effective to maintain the inference accuracy when a better control is favored. To somehow, it is a fair thing that we need to sacrifice something for the others.  But what is point of this paper instead?
3. The authors present the same set experiments with two styles, (i.e., figure and table). But these two styles carry the same results and same observation. I believe only choosing one style for result presentation is sufficient and more space can be saved for more experiments with other model sizes and manufacturers.

**Questions:**

All my questions are already discussed in Weaknesses Section.

---

> ### Author Response · Authors · 2025-12-03
>
> We thank the reviewer for the detailed comments and for raising several important concerns. Below we respond to each weakness in turn.
>
> ---
>
> **(1) On using only Qwen2.5-7B and model generality. (Weakness 1)**
>
> We agree that relying on a single backbone is not fully satisfactory. In the rebuttal revision, we have added results on **LLaMA-3.1-8B-Instruct** (see Appendix) and **Phi-4-14B** (Section 4). These additional experiments support the same qualitative findings as with Qwen2.5-7B-Instruct, indicating that our conclusions are not tied to a single model family or size.
>
> We chose Qwen2.5-7B-Instruct as our primary backbone because the Qwen2.5 series is widely used for post-training and distillation (e.g., DeepSeek-R1 distillation) and has proven to be a strong and stable base for hybrid-thinking SFT. This makes it a natural and representative testbed for analyzing the factors behind hybrid-thinking controllability.
>
> ---
>
> **(2) On the trade-off in no-think mode and the point of the paper. (Weakness 2)**
>
> We fully acknowledge the trade-off observed in the no-think mode. The point of our paper is precisely **not** to claim that this trade-off disappears, but to **characterize and improve the balance** within the hybrid-thinking paradigm and its trade-off. Concretely, we show how data scale, pairing strategy, think:no-think ratio, and two-phase training shift the model along this trade-off, and we provide a practical recipe that “costs less and obtains more”:
>
> - by sacrificing paired repetitions to gain larger effective data scale (no-pairs),
> - and by structuring the same dataset into a two-phase pipeline.
>
> In Section 4, our best recipe achieves substantially shorter and cleaner no-think outputs while maintaining competitive accuracy compared to standard hybrid-training baselines. This actionable guidance on how to train hybrid-thinking models under an inherent trade-off is exactly the main contribution of the paper.
>
> ---
>
> **(3) On duplicated figures and tables vs. more experiments. (Weakness 3)**
>
> We understand the concern that figures and tables sometimes present overlapping results. Our intention is that they serve complementary roles:
>
> - **Tables** give precise numeric values for detailed comparison.
> - **Figures** make trends (e.g., how length shrinks with data scale, or how different ratios behave) immediately visible, which is much harder to see from tables alone.
>
> We agree that more experimental coverage is important. In the rebuttal revision, we have added further results (including larger and different backbones) to the Appendix, while keeping only the most informative combinations of figures and tables in the main text. We believe this strikes a better balance between clarity of presentation and experimental breadth.
>
> ---
>
> We again thank the reviewer for the comments. We have incorporated the above clarifications, additional experiments, and wording changes into the rebuttal revision to better highlight the purpose and impact of our work.

---

### Official Review · Reviewer_wAFL · 2025-11-02

**Soundness:** 3
**Presentation:** 3
**Contribution:** 2
**Rating:** 4
**Confidence:** 4

**Summary:**

This paper analyzes the key factors influencing the controllability of hybrid thinking training via Supervised Fine-Tuning (SFT), with a specific focus on the tendency of Large Reasoning Models (LRMs) to persist in a non-thinking mode during the final answer generation. The authors substantiate four key findings through extensive experiments on the Qwen2.5-7B-Instruct model. While numerous SFT data processing methods exist to enhance reasoning efficiency, the experimental validation of the proposed method is limited. These limitations include non-comparable data scales, a constrained selection of base models, and a lack of baseline comparisons. Consequently, more comprehensive experimentation is required to firmly establish the method's efficacy

**Strengths:**

1. The paper thoroughly analyzes the key factors affecting the controllability of hybrid thinking during SFT. Extensive experiments on the math datasets with Qwen2.5-7B-Instruct model demonstrate that the efficiency of the no-think mode can be improved with larger data scale, better data ratio, non-paired data, and improved two-phase training. These findings are helpful for SFT training for efficient LRMs.

2. The new recipe effectively improves the controllability of the no-think mode, where the number of thinking steps is dramatically reduced than the original recipe on two datasets.

**Weaknesses:**

1. The ratio of think and no-think data is more a trade-off between reasoning accuracy and generation length according to Table 4. In the two-phase training part and Section 4, the data mixture ratios of think and no-think are 1:1 and 5:4, respectively. Therefore, the data ratio finding is not practically used to validate its effectiveness.

2. The main advantage of the proposed training method is the more controlled and efficient no-think mode. There are no obvious improvements for the more time-consuming and important thinking mode. The controllability of no-think mode can be improved with better prompts. Therefore, this method should also be compared with other SFT and training-free methods in the perspective of efficiency and effectiveness trade-off.

3. In addition, in Table 6, the four findings are not used to improve hybrid thinking over existing training settings. Only Qwen2.5-7B-Base and LLaMA-3.1-8B-Base are fine-tuned in Section 4, and the recent and powerful Qwen3 or  other base models are helpful to validate the generalization of the proposed method.

**Questions:**

See weaknesses.

---

> ### Author Response · Authors · 2025-12-03
>
> We thank the reviewer for the detailed feedback and for highlighting several important clarification points. Below we respond to the three main concerns.
>
> ---
>
> **(1) On the data ratio of think vs. no-think samples (Table 4). (Weakness 1)**
>
> We apologize for the confusion caused by our notation. The setting labeled as “1:1” in Table 4 actually corresponds to a think:no-think ratio of \(5:4\). We rounded it to “1:1” in the figure for visual simplicity.
>
> In the rebuttal revision, we explicitly state in the caption and main text that “1:1” denotes the \(5:4\) condition to avoid ambiguity. Thus, the ratio finding is indeed used in our final recipe: the best-performing setting combines (approx.) \(1:1\)–\(5:4\) mixtures with large-scale data and two-phase training.
>
> ---
>
> **(2) On the goal of our method and the role of think vs. no-think modes.  (Weakness 2)**
>
> The reviewer is correct that our method mainly improves the controllability and efficiency of the *no-think* mode, and that the *think* mode accuracy remains roughly unchanged. This is, in fact, the intended focus of the paper rather than a limitation of the method.
>
> Hybrid thinking, as implemented in models for example Qwen3, is defined as controlling reasoning behavior via a small set of special tokens (e.g., `\think` and `\no_think`). Our work is specifically targeted at the key unresolved failure mode of this paradigm: even under `\no_think`, models still leak long reasoning traces and reflection-style tokens (e.g., “wait”, “hmm”), but **not** improving its thinking mode.
>
> Prompt-based or more elaborate training-free methods can certainly be used to shorten answers or suppress reasoning, but they fall outside this definition of hybrid thinking which controls reasoning mode using only a speical token. Our contribution is complementary: we provide a principled SFT recipe that improves controllability **within** the existing hybrid-thinking framework, but **not** analyzing the advantage of hybrid-thinking compared to other methods.
>
> ---
>
> **(3) On the connection between the four findings and the final recipe, and on model choices. (Weakness 3)**
>
> Regarding Table 6, our final recipe is indeed derived directly from the four empirical findings in Section 3:
>
> - **Finding 1 (data scale):** We use 140k samples hybrid-thinking data.
> - **Finding 2 (pairing):** We adopt the *no-pairs* setting to increase the dataset scale.
> - **Finding 3 (data ratio):** We use the approximate \(1:1\) (i.e., \(5:4\)) think:no-think ratio.
> - **Finding 4 (two-phase training):** We first train on pure think data and then apply hybrid-thinking SFT.
>
> In the rebuttal revision, we add a short statement that explicitly states this mapping from Findings in Section3 to the concrete recipe in Section 4, so that the connection is clearer.
>
> On the backbone choice, we would like to clarify that we fine-tune **instruction-tuned** models, namely Qwen2.5-7B-Instruct and LLaMA-3.1-8B-Instruct, not the base models. These backbones are suitable for recent pose-training and distillation work (e.g., DeepSeek-R1 distillation), which is crucial for hybrid-thinking training.
>
> At the time of our experiments, we did not have access to stable, publicly released *instruction-tuned* Qwen3-Instruct models for hybrid-thinking SFT. Directly fine-tuning Qwen3-base models for hybrid thinking would mix our recipe with the yet-unquantified effects, which would be an unfair comparison. For these reasons we focused on Qwen2.5-7B-Instruct and LLaMA-3.1-8B-Instruct in this first study.
>
> ---
>
> We again thank the reviewer for the thoughtful comments. We have incorporated the above clarifications and wording changes into the rebuttal revision to make our contributions and experimental design clearer.

---

### Author Response · Authors · 2025-12-03
**Integrated Summary of the Paper and Rebuttal (1)**

Dear Area Chair,

We hope you had a pleasant Thanksgiving.

Thank you for taking on the responsibility of handling this submission under the unexpected circumstances surrounding ICLR 2026 and the OpenReview incident. We fully appreciate the additional workload this creates. To make it easier for you to quickly understand our work and our rebuttal, we have prepared this concise summary.

---

## Brief Description of the Paper

Hybrid thinking models exhibit a limitation: reasoning leakage in the "no-think" mode. To address this limitation, our paper systematically analyzes which training factors can mitigate this behavior, and then proposes a practical recipe that significantly reduces no-think leakage while preserving reasoning performance in the think mode.

## Overview of the Reviews

We received four reviews:

- Some comments are very helpful and insightful, especially the weaknesses raised by **Reviewer 3 (kdYd)**. We have added experiments and strengthened the exposition in response.
- Some comments reflect misunderstandings, not only about specific details but also about the core topic/motivation of our paper, particularly from **Reviewer 1 (wAFL)** and **Reviewer 2 (xR3d)**. We clarify these points below.
- In addition, **Reviewer 4 (PBRL)** raised a very vague remarks, where the exact target or intended meaning was unclear. Even so, we have added experiments and discussion as much as we reasonably could.

## Clarifying Key Misunderstandings

### 1. Misunderstandings about the main goal of the paper

1. **Think vs. no-think focus (Reviewer 1, wAFL).**
   The reviewer states that our method does not improve the **think** mode. This is correct, but it is not a weakness: improving the think mode is **not** the goal of this paper. Our focus is on **fixing a core deficiency of hybrid thinking in the no-think mode—reasoning leakage**—and on improving controllability and efficiency under `\no_think`.

2. **Comparison to prompt-based / training-free methods (Reviewer 1, wAFL).**
   The reviewer suggests that we should compare against prompt-design or other training-free methods for reducing reasoning leakage. However, our object of study is **hybrid thinking as currently implemented in practice** (using a small set of control tokens such as `\think` / `\no_think`) and how to **repair its limitations**, rather than to position hybrid thinking against all possible training-free alternatives.

3. **“There is a trade-off, so the paper has no point” (Reviewer 2, xR3d).**
   Reviewer 2 notes that there is an accuracy–length trade-off in the no-think mode and argues that such a trade-off is obvious, implying that the paper has no point. We **explicitly acknowledge** that hybrid thinking inherently involves trade-offs. The core aim of our work is precisely to **characterize and balance this trade-off** and to provide **practical guidance and a concrete recipe** for training hybrid-thinking models: how to “pay less and get more” in terms of no-think controllability while preserving accuracy. This is the central point of the paper.

4. **Simpler tasks for analyzing the cause of leakage (Reviewer 3, kdYd).**
   Reviewer 3 suggests adding simpler, general-purpose tasks to analyze whether reasoning leakage in the no-think mode is due to the difficulty of the tasks. Our paper does **not** aim to fully explain *why* leakage occurs or under exactly which conditions it first appears. Instead, we take leakage in hybrid thinking as an established problem and focus on **how to control and mitigate it** in the regime where large reasoning models are actually deployed: challenging math and science reasoning tasks.

### 2. Misunderstandings about specific details

1. **Data ratio “1:1” vs. “5:4” (Reviewer 1, wAFL).**
   The reviewer comments that the 1:1 and 5:4 ratios are too similar to be distinguished. We clarify that **“1:1” is simply a rounded notation for the 5:4 condition**—they are the **same** experimental group. In the revised version, we explicitly state that “1:1” denotes the 5:4 ratio.

2. **Base vs. instruct models and Qwen3 (Reviewer 1, wAFL).**
   The reviewer states that we only use Qwen2.5-7B-Base and LLaMA-3.1-8B-Base and suggests adding Qwen3-Base. In fact, our paper clearly states that we use **Qwen2.5-7B-Instruct** and **LLaMA-3.1-8B-Instruct**, not base models. At the time of our experiments, **Qwen3 did not provide a public instruction-tuned (Instruct) version**, so we did not use Qwen3 as a backbone.

3. **“Only two simple benchmarks” (Reviewer 4, PBRL).**
   Reviewer 4 claims that we only use “two simple mathematical benchmarks”. This is inaccurate. We use **four** benchmarks:
   - lower-to-moderate difficulty: **MATH500**, **MMLU-STEM**,
   - higher difficulty: **AIME24**, **GPQA-Diamond**.
   We therefore believe our task selection is reasonably diverse within the math/science reasoning domain.

**(The continuation of our summary appears in the subsequent comment)**

---

### Author Response · Authors · 2025-12-03
**Integrated Summary of the Paper and Rebuttal (2)**

**(This comment continues the summary from the previous one)**

## Improvements and Added Experiments

1. **More backbones in Section 3 (Reviewer 2, xR3d).**
   Reviewer 2 points out that analyzing hybrid-thinking training factors using only Qwen2.5-7B-Instruct is not sufficient. We agree and have added new experiments based on **LLaMA-3.1-8B-Instruct** in Section 3. The detailed results are included in the Appendix.

2. **Larger models and MoE (Reviewer 3, kdYd).**
   Reviewer 3 suggests adding experiments on larger models or MoE models. We have added experiments on **Phi-4-14B**, a larger dense model, and observe qualitatively similar phenomena and gains from our recipe.
   Regarding MoE, recent work shows that **under comparable training paradigms and resource budgets, MoE and dense models exhibit similar downstream behavior during SFT**. Thus, we do not believe that MoE experiments are essential for supporting our main claims in this paper.

3. **“Generalization of this method on no-think mode.” (Reviewer 4, PBRL).**
   This comment is very vague; it is not clear which aspect of “generalization” is being requested (across tasks, domains, or architectures). Nevertheless, we have strengthened the evidence for generalization by:
   - adding **LLaMA-3.1-8B-Instruct** experiments for the factor analysis in Section 3, and
   - adding a larger model, **Phi-4-14B**, in Section 4.
   We believe these additional results are sufficient to show that our findings and recipe generalize across multiple dense backbones and model sizes in the no-think setting.

---

We hope this summary helps you quickly grasp the main ideas of our work and how we responded to the reviewers’ concerns. For full details, please refer to our point-by-point rebuttals to each reviewer.

Thank you very much for your time and consideration, and we wish you a pleasant December.

---

### Note · Authors · 2026-01-05

I have read and agree with the venue's withdrawal policy on behalf of myself and my co-authors.